# Time to Classify Tumours of the Stomach and the Kidneys According to Cell of Origin

**DOI:** 10.3390/ijms222413386

**Published:** 2021-12-13

**Authors:** Helge Waldum, Patricia Mjønes

**Affiliations:** 1Department of Clinical and Molecular Medicine, Faculty of Medicine and Health Sciences, Norwegian University of Science and Technology, 7030 Trondheim, Norway; patricia.mjones@ntnu.no; 2Department of Pathology, St. Olav’s Hospital, 7030 Trondheim, Norway

**Keywords:** cell of origin of tumours, tumour classification, neuroendocrine cells, neuroendocrine carcinoma versus adenocarcinoma, gastric carcinoma, renal carcinoma

## Abstract

Malignant tumours are traditionally classified according to their organ of origin and whether they are of epithelial (carcinomas) or mesenchymal (sarcomas) origin. By histological appearance the site of origin may often be confirmed. Using same treatment for tumours from the same organ is rational only when there is no principal heterogeneity between the tumours of that organ. Organ tumour heterogeneity is typical for the lungs with small cell and non-small cell tumours, for the kidneys where clear cell renal carcinoma (CCRCC) is the dominating type among other subgroups, and in the stomach with adenocarcinomas of intestinal and diffuse types. In addition, a separate type of neuroendocrine tumours (NETs) is found in most organs. Every cell type able to divide may develop into a tumour, and the different subtypes most often reflect different cell origin. In this article the focus is on the cells of origin in tumours arising in the stomach and kidneys and the close relationship between normal neuroendocrine cells and NETs. Furthermore, that the erythropoietin producing cell may be the cell of origin of CCRCC (a cancer with many similarities to NETs), and that gastric carcinomas of diffuse type may originate from the ECL cell, whereas the endodermal stem cell most probably gives rise to cancers of intestinal type.

## 1. Introduction

Tumours were initially classified macroscopically according to the organ where they appeared. After the development of the histological technique including fixation of tissue followed by making thin slices allowing different staining of tissue components, the classification could also rely on histology. Tumours were early recognized to spread from one organ to another by metastasis, and in such situations histological differences between the tumours became useful to determine their organ of origin. This organ classification of neoplasia still prevails and is used to select treatment. From a biological point of view this may be a rational approach if there is no heterogeneity in cellular origin of tumours from the actual organ. However, this is not the case, since many cell types in an organ are able to divide and thus give rise to tumours. In the present review we will discuss tumour classification with respect to tumour biology, including the cells of origin.

## 2. Tumour Biology

In a multicellular organism the mass of the different cell types is tightly controlled. The concept of chalones representing molecules that were produced by a certain cell type and inhibiting the proliferation of the same cell type, was popular about 50 years ago [1]. It has not maintained this popularity later because chalones were not identified. However, the concept is nevertheless intellectually attractive since it explains what is generally recognized, the constancy of different cell masses. It may also be that the search for chalones has been unsuccessful because the wrong molecules have been looked for. Perhaps molecules having an important signal function on other types of cells could have a negative trophic effect on the producing cell itself. For instance, on endocrine cells there is often a receptor for the signal substance on the same cell [2]. If this receptor were to transmit a positive trophic effect, a positive autocrine mechanism in tumorigenesis would exist [3]. From a biological point of view, it is much more probable that this auto-receptor should transmit negative functional and trophic effects [4] and thus the ligand acts as a chalone.

The mass of the different cell types is regulated according to the need or use of its function. This is easily recognized macroscopically in persons performing heavy muscular work. However, when changing to a sedentary life, the muscle mass returns gradually to a lower level. The increase in muscle mass secondary to heavy work or training may be due to neural factors parallel to the stimulation of muscle use. In the stomach it has been convincingly shown that the vagal nerves not only stimulate acid secretion, but also have a positive trophic effect [5]. Generally, there is a close correlation between regulation of function and proliferation as demonstrated for the enterochromaffin like (ECL) cell where gastrin stimulates histamine release as well as proliferation [6]. The hormones are the most important trophic factors. They change the “set-point” for their target cells, increasing it in cells where the hormone stimulates the function and reducing it when the stimulation is removed [7] or where the hormone has an inhibitory effect on function like somatostatin [8]. For at least peptide hormones the effects on function and proliferation are mediated via the same membrane receptor which is maximally stimulated when the actual hormone is bound to all its receptors. Thus, hormone concentrations exceeding the level where all receptors are saturated, will have no additional effect. Against this background it is strange that, as previously claimed, only very high gastrin values would play any role in tumorigenesis, and that gastrin elevation secondary to proton pump inhibitor (PPI) treatment would not reach values of concern if anacidity was avoided [9]. On the contrary, there is no lower limit for the trophic effect which is exerted through the normal range as well [10,11]. Similarly, removal of the antrum to reduce acid secretion via hypogastrinemia, which a period was used in the treatment of peptic ulcer disease, resulted in oxyntic atrophy, and at least in the rat a particular reduction in ECL cells [12]. Patients operated with an antrectomy have a risk of developing cancer in the remaining part of the stomach, so-called stump carcinoma. Since gastrin has a negative trophic effect on the D (somatostatin producing) cells at most locations, we hypothesized that the postoperative hypogastrinemia via reduction in growth inhibition of D cells could predispose to stump cancers. We found expression of the neuroendocrine marker neuron-specific enolase (NSE) in a third of the stump carcinomas, and among them one with somatostatin expression [13]. Otherwise, we only know of one D cell-derived tumour, a highly malignant cancer localized to the oxyntic mucosa, expressing somatostatin [14]. A continuous trophic overstimulation will not lead to a steady increase in the density of hyper stimulated cells since the increase in cell mass necessarily will augment the release of substance(s) inhibiting their own proliferation (functional chalones), and a new set point will be established preventing further increases. Nevertheless, there will be hyperplasia of the target cell which through the forced and augmented proliferation is at increased risk of mutations and possibly other genetic changes. Mutations even where growth regulation is only slightly changed, may manifest themselves by forming polyps. Such polyps are apparently reversible as they often disappear when the hormonal overstimulation is removed [15,16]. Whether all the mutated cells are gone after regress of macroscopical polyps, is, however, uncertain. The long-term tumorigenic effect of hormones is thus due to the ability to induce genetic changes (initiation) and then stimulate the mutated cells further to proliferation (promotion and progression). Thereby hormones become complete carcinogens. This was shown more than 50 years ago for sex hormones when girls borne of mothers having been treated with oestrogens during pregnancy, developed cancers of the vagina at an early age [17]. Similarly, and more recently also for gastric cancer that developed from ECL cells in young adults due to hypergastrinemia caused by missense mutation of one of the genes coding for the proton pump (Figure 1) [18].

The central role of hormones in carcinogenesis is also clearly shown by the difference in prevalence of breast cancer between the two sexes.

## 3. Cell of Origin of Tumours

Examinations of tumours to detect mutations to tailor treatment have given rise to great expectations during recent years. Although some individual improvement in treatment may be gained by such an approach, this will not be of much help for understanding the etiology and pathogenesis of the tumour since by definition “spontaneous” mutations occur by chance. Instead of going directly from tumour classification based just on the organ to individual genetic changes in tumours, the rational approach would be to determine the cell of origin. Based upon the knowledge of its regulation, new information of the tumour pathogenesis will be gained. Examination for receptors normally present on the cell of origin, will then give information on the possibility to use hormone antagonists or agonists in the treatment. For example, our demonstration of the central role of the erythropoietin producing cell in CCRCC (clear cell renal cancer cell) [20] explains the recently detected effect of HIF (hypoxia inducible factor) antagonist in the treatment of such carcinomas [21]. Cell of origin is most easily determined by immunohistochemistry or in-situ hybridization although also whole genome sequencing is a possibility. However, in the latter case selection of single actual candidate cells for comparison may be a problem.

Tumours with traits of epithelial cells and having developed from such cells are classified as carcinomas, whereas those without such features generally are called sarcomas. There have been two theories for the pathogenesis of malignant neoplasia, dedifferentiation of normal cells [22] and the other, stop in differentiation of a primitive cell like a stem cell [23]. It is, however, evident that only cells able to divide may give origin to a tumour. Probably, both dedifferentiation of mature cells and stop in differentiation of immature cells both give rise to tumours.

## 4. Neuroendocrine Cells

Neuroendocrine (NE) cells occur in most external surfaces, and they can divide [24], although some have claimed that NE cells do not divide in man [25]. However, they do proliferate very slowly, which might explain the disputes about their ability to divide. The slow normal proliferation is also reflected in the NETs, although being malignant with the ability to metastasize, they grow slowly which explains the often long-survival of NET patients. Moreover, no chromogranin A positive cells were shown to divide in gastric carcinomas, and therefore chromogranin A positive tumour cells were claimed to be quiescent cells without any role in tumorigenesis [26]. Alternatively, this phenomenon may be explained by lack of chromogranin A expression during mitoses. The tumour cells in NETs are phenotypically very similar to the normal NE cell of origin, which probably reflects the low mutation rate in these tumours [27,28]. In contrast to other epithelial cells, normal NE cells do not adhere to each other, possibly due to reduced expression of adherence molecules. In fact, we could not detect E-cadherin on the ECL cell in the oxyntic mucosa [29] suggesting that this cell was prone to develop into a tumour. Another consequence of accepting that differentiated cells may degenerate into a tumour is the possible role of normal signal substances in tumorigenesis. Thus, there are about 20 different NE cells in the gastrointestinal tract, but mainly enterochromaffin (EC) and ECL cells [30] develop into tumours although D cells may rarely give rise to tumours [31] and even more seldom A-like (ghrelin producing) cells [32]. The different NE cell types are very similar both anatomically and functionally, why then this great discrepancy in tumorigenesis? The high prevalence of atrophic oxyntic gastritis leading to reduced gastric acidity and hypergastrinemia, is of course an important factor for ECL cell tumorigenesis. The knowledge of the functional and proliferative regulation of the EC cell is much less [33]. However, the EC and ECL cell have another property in common, the production of vasoactive signal substances, serotonin, and histamine, respectively. Histamine through its vascular dilatation and permeation may facilitate the spread of tumour cells, and together with another ECL cell mediator, REG(regenerating) protein, have a trophic effect on another cell, the stem cell [34]. Serotonin has also profound vascular effects, but probably more importantly, a stimulatory effect on fibrosis which manifests itself in the proximity of EC cell NETs as tumour desmoplasia [35,36]. The effects of serotonin are mediated by multiple different receptors and its main functions seem to be as neurotransmitter. Outside the central nervous system, the EC cell is the main producer of serotonin [37]. It is taken up by megakaryocytes and platelets and transported by the latter to the place of need. As other platelet constituents, serotonin is released from the platelets upon blood sampling as well as separation of blood corpuscles from plasma. This makes assessment of serotonin in blood very difficult, and it is possible that free serotonin does not circulate in blood during normal conditions although it is detectable in patients with EC cell NETs [38]. In other words, is serotonin a real hormone, or just a neurotransmitter and a signal substance transported by the platelets? In any way, serotonin alone can induce valvular heart disease as exemplified in patients with EC cell NETs metastasized to the liver [39] or in rats dosed long-term with serotonin [40]. NETs are often divided into those giving hormonal overproduction syndrome like serotonin or histamine flush, and those without such syndromes. It has, however, to be realized that this does not necessarily reflect important differences between the tumour cells. The presence of hormonal overproduction syndromes depends on the acute effects of the signal substance, for instance insulin, and the site of release as EC cell NETs give rise to serotonin flush even before liver metastasis when the primary is localized to the lungs. The peculiarities of NETs are summarized in Table 1.

## 5. Tumorigenesis Is Due to Genetic Changes

Only a low percentage of tumours is due to congenital genetic changes [41,42]. However, tumours develop due to acquired changes in genes, most often mutations. Without mutations there would have been no evolution, but at the same time it is the principal mechanism in the development of neoplasia. Only a low proportion of mutations results in a functional improvement which is preserved and contributing to the evolution. Some mutations affect silent parts of the gene and have no functional impact. However, most mutations have negative effect on the function of the gene product. Therefore, the apparent gain of function in tumour cells like increased proliferation and invasiveness represent a loss of function of a regulatory mechanism. Thus, very few new molecules are expressed in cancer cells compared with the normal cell, but during carcinogenesis markers are gradually lost which may make cell of origin difficult to identify [43]. However, when a compound relatively selectively is produced in a cell or a cell type, assessment of this compound may be used as a marker for such cells. Chromogranin A is a typical example of such a compound being specific for NE cells as it is found only in secretory granules [44] and accordingly increased in persons with NE cell hyperplasia [45] and NE tumours of low malignancy [45]. With increased malignancy the tumour cells lose specific traits including expression of secretory granules, and chromogranin A in blood may fall despite increased mass of tumour cells. In a study from our group, we found blood chromogranin A elevation in six adenocarcinomas. Five of these tumours expressed chromogranin A in their tumour cells when examined with immunohistochemistry with increased sensitivity by using tyramide signal amplification, which may suggest misclassification of neuroendocrine carcinomas as adenocarcinomas [46].

## 6. Specificity of Cellular Markers

From what is written above it seems unlikely that a new marker may appear in a malignant cell among other cells not expressing this marker. Thus, using a certain number for the percentage of tumour cells expressing a neuroendocrine marker to differentiate between a neuroendocrine cancer and an adenocarcinoma [47], seems illogical. The rational approach is to use the most sensitive and specific method available for the detection of the marker. We found expression of the neuroendocrine marker chromogranin A in gastric carcinomas in patients with pernicious anaemia and initially classified as adenocarcinomas. By immunohistochemistry with tyramide signal amplification (TSA) the tumour expressed neuroendocrine markers [48]. These neuroendocrine carcinomas are an example of a gradual development of increasing malignancy of ECL cell derived tumours due to hypergastrinemia. In an individual patient with pernicious anaemia we observed how a removed gastric NET of ECL cell origin recurred after 5 years as a highly malignant neuroendocrine carcinoma (NEC) that killed the patient [49]. In contrast to histochemical methods, immunohistochemistry and in-situ hybridization methods are much more specific and thus more reliable [50]. For example, Periodic Acid-Schiff (PAS) positivity has been thought to be rather specific for mucin and thus to reflect exocrine origin of tumour cells. This belief has been the sole reason for classifying gastric cancers of diffuse type among adenocarcinomas [51]. However, PAS binds to glycoproteins in general, a fact that has been known for decades [52]. Accordingly, there is no reason to classify gastric carcinomas of diffuse type as adenocarcinomas, but instead as neuroendocrine carcinomas since they disclose neuroendocrine markers by immunohistochemistry [48,49,53,54,55,56], in-situ-hybridization [56,57] and by immuno-electron-microscopy [58]. The specificity of immunohistochemistry is not absolute, particularly when using polyclonal antibodies which may contain antibodies directed towards many different epitopes which may be shared by many antigens. Monoclonal antibodies, of course, are directed towards a single epitope, but even an epitope may be found in closely related antigens [59]. The loss of specific traits like the occurrence of phenotypically normal secretory granules during the development of malignancy, may make correct identification of cell of origin difficult [49,60]. The best specificity is obtained by in-situ hybridization allowing selection of specific DNA/RNA sequences [61] as exemplified for Epstein-Barr virus [62] and recently COVID-19 virus [63]. We have had very good experience with the commercially available ISH kit RNA-scope (Advanced Cell Diagnostics, Newark, CA 94560, USA) where multiple probes hybridizing in pairs followed by an amplification step, are applied [61]. By paired hybridization the specificity is a priori improved. In-situ hybridization which is time-consuming should not be used in routine pathological examinations, but instead used to control other less specific methods in every-day use. The specificity of a marker may also be underestimated as we showed for neuron specific enolase (NSE) which had been claimed to be rather unspecific although NSE positive tumours have been shown to be increasingly positive for other neuroendocrine (NE) markers when increasing the sensitivity of the methods and/or increasing the number of NE markers used [64].

## 7. Diagnostic Consequences of Improved Methods in Pathology

To improve understanding, prophylaxis, and treatment of neoplasia a thorough and detailed pathological classification of neoplasia is very important. Using specific markers with the highest sensitive method to identify the cell of origin will induce a more biologically oriented tumour classification which will be helpful in treatment at all phases of neoplasia. Thus, substances stimulating a certain cell to proliferation will necessarily increase the risk of tumour development, and similarly, compounds blocking this effect will reduce tumour growth if the receptors are maintained (Figure 2). Signal substances from nerves and endocrine cells (hormones) are the most important regulators of growth. The importance of identifying the cell of origin as well as its main driver have been realized for lung cancers [65,66], a cancer where neuroendocrine cell types make up an important proportion of the different types. Interestingly, tuft cells have been claimed to be related to a proportion of small cell cancers devoid of secretory granules [67]. Subsequently, we will discuss the cell of origin in more detail for gastric and renal cancers, two localizations our group has studied with respect to cell of origin.

## 8. Gastric Cancer

Gastric cancer was previously very prevalent and combined with a bad prognosis, gastric cancer was one of the deadliest types of cancer. During the last decades there has been a remarkable reduction in gastric cancer prevalence [68,69]. Laurén classified gastric carcinomas into intestinal type (those showing glandular growth pattern) and diffuse type (those showing no glandular growth) and a third group of about 15% showing features of the two other groups [51]. The diffuse type was also considered to be adenocarcinomas due to its positive reaction to PAS, a stain regarded as a specific marker of the exocrine substance mucin, which, however, is not correct [52]. The fact that the decline in prevalence is mainly due to a reduction in the occurrence of the intestinal type [19,70], strongly suggests different etiology of the two types. It was also early recognized that one type seldom transforms into the other in line with different etiology and pathogenesis.

It was also realized that gastric carcinomas seldom develop in a stomach without gastritis [71], an observation confirmed many years later with the description of *Helicobacter* as the cause of gastritis [72] as well as an important proportion of gastric cancers [73]. Never had a bacterial infection been shown to cause a malignant tumour. However, despite immense effort, no factor in *Helicobacter pylori* has been shown to induce neoplasia, nor any specific susceptibility in persons who developed gastric cancer secondary to *Helicobacter pylori* infection has been disclosed. On the other hand, a very important step in the understanding of the carcinogenic mechanism was the description that *Helicobacter pylori* gastritis predisposed to gastric cancer only after having induced oxyntic atrophy [74]. This condition is accompanied by reduced acid secretion, reduced gastric acidity and thus hypergastrinemia suggesting that hypergastrinemia could mediate the pathogenesis of gastric cancer [75]. The fact that the predisposition to gastric cancer persists after eradication of *Helicobacter pylori* if atrophic gastritis has already developed [76] also strongly indicates that the carcinogenic effect is an indirect one. *Helicobacter pylori* gastritis, however, predisposes to both gastric cancer of diffuse and intestinal types [77], although generally the detection of *Helicobacter pylori* has been higher in the intestinal type compared with the diffuse type [77]. Based upon histochemistry [53], immunohistochemistry [54], immuno-electron microscopy [59] and in-situ hybridization [56,57] we have repeatedly shown that ECL cell markers are mainly expressed in carcinomas of diffuse type, and therefore we have concluded that these carcinomas originate from the ECL cell [53,54,78]. On the other hand, the intestinal type of gastric cancer may develop from the stem cell [78] where gastrin also stimulates proliferation either by a possible gastrin receptor or via Reg released from the ECL cell [79] (Figure 3 and Figure 4). For the understanding as well as treatment of tumours it is essential to identify the cell of origin [80] as knowledge of its receptors and growth regulation give information of tumour pathogenesis and possible treatment options. Antagonists of receptors transmitting stimulation of proliferation and agonists of receptors signaling negative trophic effect will be expected to have an inhibitory effect on tumour growth during stages of tumorigenesis. For gastric cancer it would accordingly be logical to treat tumours of diffuse type as being of ECL cell origin and with intact gastrin receptor, with a gastrin antagonist like netazepide [81,82]. In carcinomas of intestinal type (stem cell origin) it is rational to use a gastrin antagonist to reduce release of ECL cell mediators including Reg protein and if available a Reg antagonist. A combination of antagonists/agonists having negative trophic effect on the proliferation of the cell of origin will be a rational strategy in the treatment. To be noticed, an intact gastrin receptor is apparently conserved during the carcinogenic process in most of the tumours [56].

## 9. Renal Cancer

Renal cell cancers are the dominant malignancies in the kidney, and among them the CCRCC is the most prevalent, making up 75% of all renal cell carcinomas [83]. Based upon the occurrence of polycythemia in about 5% of patients with CCRCC we examined renal carcinomas for erythropoietin expression in tumour cells, and found it present in all tumours [84]. Furthermore, there are some clinical similarities with neuroendocrine tumours (NETs) like relatively slow growth but early metastasis together with the erythropoietin secretion in a percentage of the cases (a parallel to hormonal overproduction in NETs). We therefore also examined the tumours for neuroendocrine expression. Neuron specific enolase (NSE) was detected in virtually all CCRCC tumours [20]. Synaptophysin was expressed in 9% of these tumours, whereas chromogranin A was negative in all. We concluded our study that CCRCC probably originated from the renal erythropoietin producing (REP) cell. However, and curiously, at least until recently, the normal REP cell was not identified with certainty, but during the last years renal cortex peritubular cells have been accepted as the REP cell [84,85]. There is indication that these pericytes express PDGFR-type-β (platelets derived growth factor receptor) [86] Nevertheless., and interestingly on the background of NSE expression in CCRCC, are the features of fibroblasts, pericytes and neurons in REP cells [86]. Erythropoietin and its receptor have also been shown to be expressed in the central nervous system [87]. A link between REP and the neuroendocrine cells is also indicated by the hereditary syndrome of phaeochromocytoma/paraganglioma and polycythemia [88].

Inactivation mutation of Von Hippel Lindau gene has for long been known to predispose to CCRCC via HIF-2 activation which subsequently stimulates the proliferation of the susceptible cell which logically is the REP cell (Figure 4) This knowledge has led to the introduction of HIF-2 antagonists in the treatment of CCRCC [89]. Moreover, recently HIF-stabilizers are taken in use in the treatment of renal anaemia [90]. The expression of erythropoietin markers in the tumour cell in CCRCC as well as the involvement of HIF-2 in the carcinogenesis and the efficiency of HIF-2 antagonists in the treatment underscore the importance of identification of the cell of origin.

## 10. Conclusions

Organs consist of different cell types which may have developed from different cells of origin including different embryological origin. Since all cells able to divide may theoretically develop into a tumour, the carcinogenic process within the same organ may differ markedly. Except for congenital mutations which are a rather rare cause of cancers, the other mutations in cancers have mostly occurred by chance, and therefore mutation analysis of tumours seldom gives any information about the cells of origin. Studies on gastric physiology have clarified the normal regulation of function and growth of the oxyntic mucosa with gastrin and its target cell, the ECL cell having a central position. Based on the expression of neuroendocrine/ECL cell markers in contrast to exocrine markers in carcinomas of diffuse type, it is reasonable to reclassify these cancers from adenocarcinomas to neuroendocrine carcinomas. We have shown how identification of cell of origin may improve understanding of the carcinogenic process and explain new treatment possibilities for gastric cancer and explain the efficacy of HIF-2 antagonists in renal cancer. Perhaps, NETs demonstrate best the importance of the cell of origin in describing the behavior of the derived tumours.

## Figures and Tables

**Figure 1 ijms-22-13386-f001:**
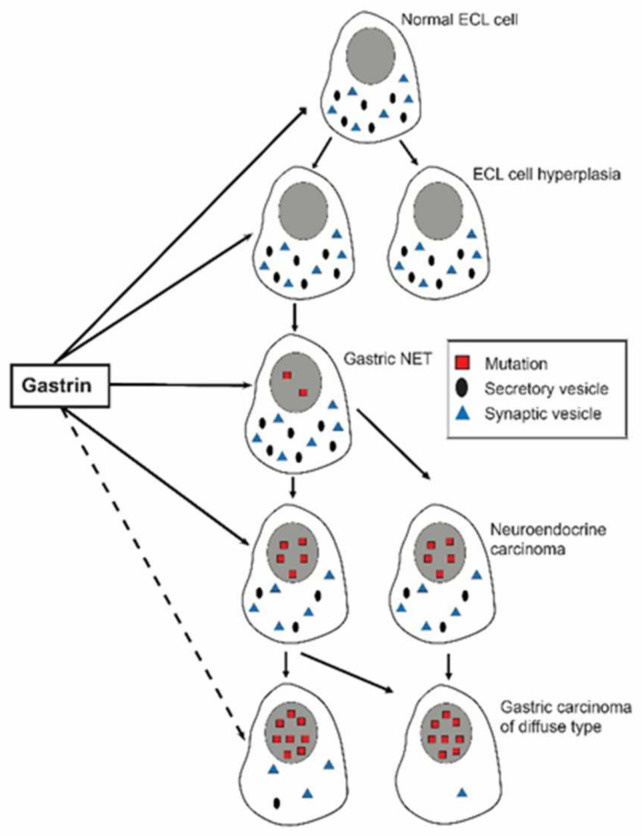
Upon chronic gastrin overstimulation the ECL cell gradually loses specific traits like secretory granules as well as receptors (With permission from ref [19]).

**Figure 2 ijms-22-13386-f002:**
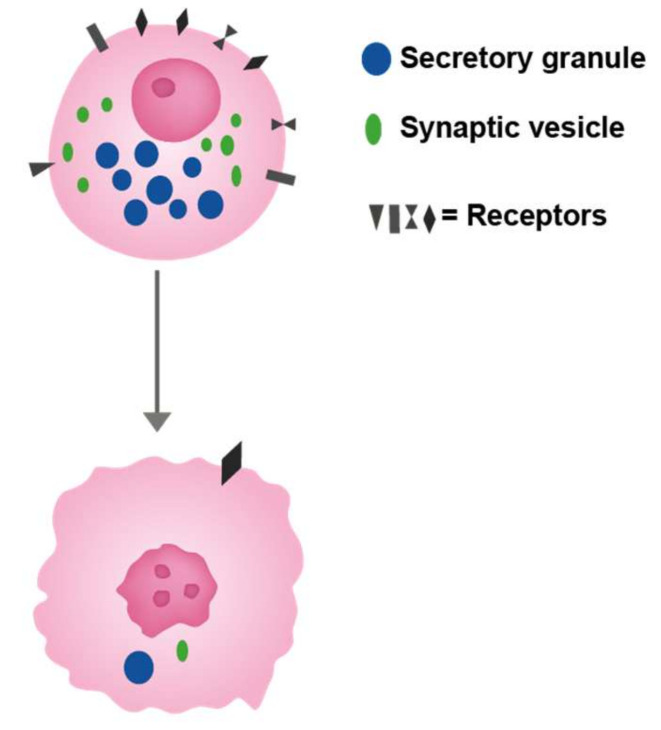
The cell of origin with different receptors regulating its growth as well as function. This cell is depicted as a neuroendocrine cell with secretory granules (upper cell). A malignant cell developed from this cell (lower cell) has lost most of the specific traits like typical secretory granules which nevertheless may be identified as that by immuno-electron microscopy. Moreover, the number of receptors is reduced, but some receptors are maintained giving rise to possible treatment option.

**Figure 3 ijms-22-13386-f003:**
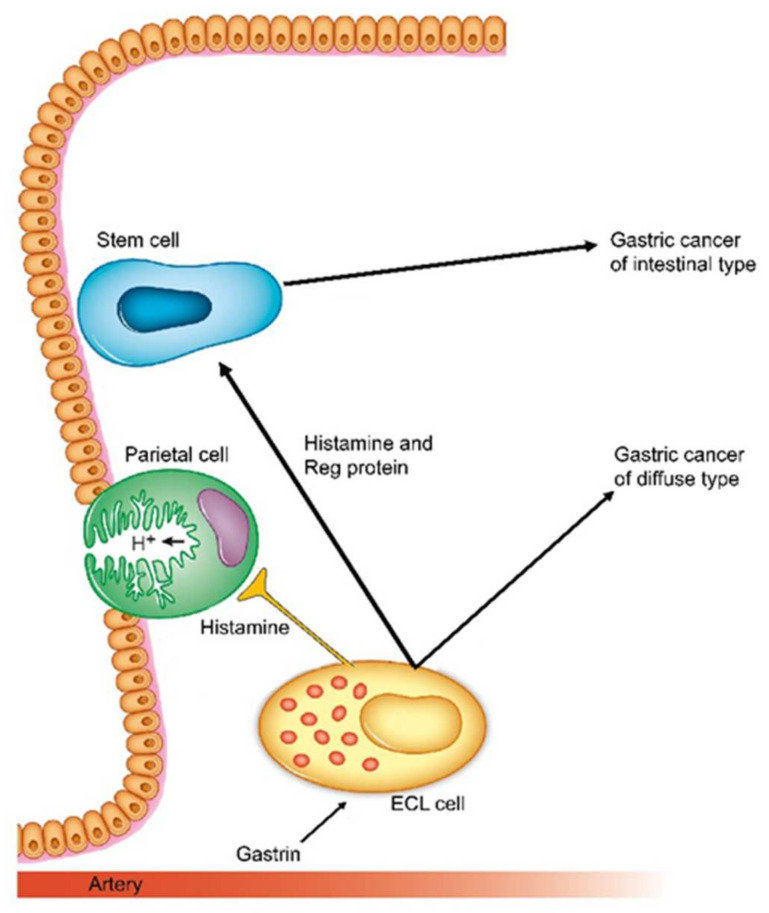
Schematic presentation of how gastrin and its target cell, the ECL cell are central in carcinogenesis both for intestinal and diffuse types (With permission from ref [78]).

**Figure 4 ijms-22-13386-f004:**
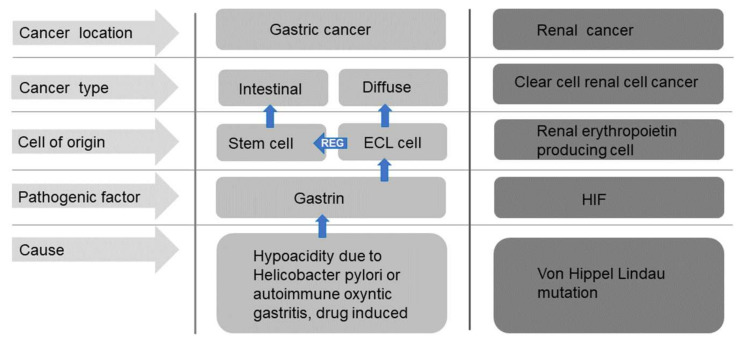
Schematic overview of the cells of origin and pathogenesis of the most important gastric and renal carcinomas.

**Table 1 ijms-22-13386-t001:** The traits of the neuroendocrine tumours reflect closely the properties of the normal neuroendocrine cells which in many respects are predisposed to tumour development.

Peculiarities with Neuroendocrine Tumours	These Peculiarities May Be Explained by Properties of the Cell of Origin
Despite an apparent benign phenotype, metastasize early	They are spread among other cells reflecting low adherence. The enterochromaffin like (ECL) cell in the stomach has been shown not to express E-cadherin. Low adherence facilitates spread.
Grow slowly. Cytotoxic drugs mainly without any effect on tumour growth	NE cells do divide, but very slowly
Tumour cells look very similar to the normal cell of origin. Low mutation rate. Immunotherapy does not seem promising	
Produce signal substances which may give symptoms if reaching the circulation in sufficient concentration and having an easily recognized effect	Produce signal substances

## Data Availability

Not applicable.

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
