# Peer review of "Time to Classify Tumours of the Stomach and the Kidneys According to Cell of Origin"

_ijms, 2021, doi:10.3390/ijms222413386_

Round 1

Reviewer 1 Report

The authors Waldum et.al have focused on the classification of tumors on the basis of their origin. They have mainly focused on cell of origin in tumours arising in the stomach and 17 kidneys, the close relationship between normal neuroendocrine cells and NETs.

The  article is well written and covered almost all teh important topics and points but i feel that teh manuscript can be improved better by adding figures summarising different facts of classification in different cancers.

Author Response

Thank you for positive comments. We have added a new figure.

Reviewer 2 Report

Waldum and colleagues aims to provide evidence about how identification of cell of origin may improve understanding of the carcinogenic process. This is a topic of interest however, I had issues to understand the manuscript.

The text is verbose, sentences are very long, punctuation marks were either put randomly or missing. In addition, the logical connection between sentences is frequently lost.

In addition, the concept by which genetic tumour profiling did not allow to capture the cell of orgin and that only cells able to divide may give orgin to tumour seems to be anachronistic. Note that, in the last 20 years, the huge amount of information captured  by whole exome sequencing studies have allowed to trace the evolution of tumour cells from their putative origin.

Author Response

We have reduced the length of some sentences by dividing them into shorter ones.. Furthermore, we have added a sentence about whole genome sequencing. On the other hand, we have not added anything about non-dividing cells giving rise to tumours as we suppose that the referee thought of apparently non-dividing cancer stem cells in carcinomas which in reality are only temporarily quiescent.

Reviewer 3 Report

The manuscript entitled “Time to Classify Tumours According to Cell of Origin” describes cell of origin in tumours arising in the stomach and kidneys. Although manuscript is written well, it has some shortcomings as follows:

Specific comments

  1. As manuscript is described on stomach and kidney cancer, author should modify the title accordingly. The present title is more generalized.
  2. In the beginning authors are focused but in the middle of the manuscript authors appear to be deviated from the main goal.
  3. Paragraphs are too long that need to be broken.
  4. Please write what ‘ECL’ and ‘PPI’ stands for.
  5. Describe briefly ‘D cells’ and ‘A-like cells’ as mentioned in page 3.
  6. There are some spelling errors such as ‘int’ (page 3), ‘isulin’ (page 4), ‘correctn’ (page 6) etc.

Author Response

  1. The title is changed according to suggestion
  2. The focus may have deviated a little in the middle of the manuscript, perhaps because we added the section about neuroendocrine cells in the later part of writing. We nevertheless feel that it is closely linked, and we prefer to keep it.
  3. :Long sentences are modified.
  4. PPIs and ECL cells are defined
  5. D and A like cells are defined
  6. Spelling errors are corrected.

Round 2

Reviewer 1 Report

The authors waldum et.al focussed on the cell of origin in tumours arising in the stomach and kidneys and the close relationship between normal neuroendocrine cells and NETs. They have discussed the resasons for classsifying the cells based on their origins. Overall teh m anuscript covers all the valid points. However I felt that the manuscript lacks proper figures to explain or convey the messages to wider auidence. Adding more figures will help teh article reach to wider audience.

Author Response

2 new figures added

Reviewer 2 Report

In this revised version of their manuscript, Waldum and colleagues improve the clarity of their manuscript.

Author Response

Thank you for your positive comments